# ContriMix: Scalable stain color augmentation for domain generalization without domain labels in digital pathology

**Tan H. Nguyen**                                     HUUTAN86@GMAIL.COM
**Dinkar Juyal**                                  DINKAR.JUYAL@PATHAI.COM
**Jin Li***                                          JINLI17255@GMAIL.COM
*PathAI Inc*
*Boston, MA 02215, USA*

**Aaditya Prakash**                          ADI.PRAKASH.ML@GMAIL.COM
*Spring Discovery Inc*
*University of California*
*San Carlos, CA 94070, USA*

**Shima Nofallah**                          SHIMA.NOFALLAH@PATHAI.COM
**Chintan Shah**                            CHINTAN.SHAHL@PATHAI.COM
**Sai Chowdary Gullapally***            SAICHOWDARYGULLAPALLY@GMAIL.COM
**Limin Yu***                                    JUSTGENEIT@GMAIL.COM
**Michael Griffin**                        MICHAEL.GRIFFIN@PATHAI.COM
**Anand Sampat***                               ME@ANANDSAMPAT.COM
**John Abel**                                  JOHN.ABEL@PATHAI.COM
**Justin Lee***                                      JLEE08@GMAIL.COM
**Amaro Taylor-Weiner***                       AMAROTAYLOR@GMAIL.COM
*PathAI Inc*
*Boston, MA 02215, USA*
*\* Employee of PathAI at the time of study*

**Editor:**

## Abstract

Differences in staining and imaging procedures can cause significant color variations in histopathology images, leading to poor generalization when deploying deep-learning models trained from a different data source. Various color augmentation methods have been proposed to generate synthetic images during training to make models more robust, eliminating the need for stain normalization during test time. Many color augmentation methods leverage domain labels to generate synthetic images. This approach causes three significant challenges to scaling such a model. Firstly, incorporating data from a new domain into deep-learning models trained on existing domain labels is not straightforward. Secondly, dependency on domain labels prevents the use of pathology images without domain labels to improve model performance. Finally, implementation of these methods becomes complicated when multiple domain labels (e.g., patient identification, medical center, etc) are associated with a single image. We introduce ContriMix, a novel domain label free stain color augmentation method based on DRIT++, a style-transfer method. ContriMix leverages sample stain color variation within a training minibatch and random mixing to extract content and attribute information from pathology images. This information can be used by a trained ContriMix model to create synthetic

images to improve the performance of existing classifiers. ContriMix outperforms competing methods on the Camelyon17-WILDS dataset. Its performance is consistent across different slides in the test set while being robust to the color variation from rare substances in pathology images. Our source code and pre-trained checkpoints are available at `https://gitlab.com/huutan86/intraminibatch_permutation_drit`.

**Keywords:** Synthetic data, Domain Generalization, Digital Pathology

# 1 Introduction

Recent advancements in Machine Learning and slide digitization have transformed digital pathology by offering high-throughput, accurate analysis on large whole-slide images (WSI) Madabhushi and Lee (2016); Ehteshami Bejnordi et al. (2017). However, pathology images often have large color variations across different labs and even within the same lab. This variation leads to poor performance of algorithms developed on certain domains (labs, scanners) when deployed on others.

Stain color normalization is often used to align the distribution of stain color of the test set to that of the training set. One way to do this is by extracting the color vectors of each stain from both sets, either from raw pixels Ruifrok et al. (2001), using Singular Value Decomposition in Optical Density space Macenko et al. (2009), or Non-negative Matrix Factorization Vahadane et al. (2016). Additionally, style-transfer methods have been proposed to perform stain normalization, leveraging frameworks such as *pix2pix* Salehi and Chalechale (2020), StainGAN Shaban et al. (2019), StainNet Kang et al. (2021), and contrastive unpaired translation Gutiérrez Pérez et al. (2022).

Stain color augmentation is another method to address the generalization problem, which can lead to better performance than stain normalization Tellez et al. (2019a). Augmentation generates several variations of input images with the same content but varied coloring to encourage the network to learn color-invariant features Tellez et al. (2019b). These methods can be divided into two groups.

Most color augmentation methods in the first group rely on domain labels. For example, HistAuGAN Wagner et al. (2021), an application of DRIT++ to pathology, learns a one-to-many mapping based on disentangling the domain-invariant content (tissue morphology) in each image from the stain color attribute of each domain. Recently, Khamankar et al Khamankar et al. (2023) suggested using adaptive instance normalization to create style-augmented synthetic images by mixing the style feature statistics of different images. These methods are dependent on domain labels and need to be retrained with every new domain, making scaling to new domains challenging. Additionally, these methods cannot take advantage of a large volume of unlabeled histopathology data to improve model performance.

Without domain labels, one way to do color augmentation is to leverage the stain color vector extraction Macenko et al. (2009); Vahadane et al. (2016) to extract stain vectors in Hematoxylin-Eosin (H&E) images and use them to generate synthetic images by color transfer for training. Recently, deep-learning methods like STRAP Yamashita et al. (2021) use style transfer to synthesize images with styles from medically irrelevant images while preserving the original high-level semantic content of pathology images.

We propose a novel color augmentation technique, ContriMix, an improvement over DRIT++ that does not require any domain labels. Like DRIT++, ContriMix disentangles the content of a pathology image (tissue morphology) from the stain color attributes (style).

Table 1: ContriMix vs other stain augmentation methods

| Method | Deep-learning based method | Doesn't need test data at training time | Doesn't need domain labels at training time | Doesn't need retraining when new domain added |
|---|---|---|---|---|
| Stain color normalization | ✗ | ✓ | ✓ | ✓ |
| StainGAN | ✓ | ✗ | ✗ | ✗ |
| DRIT++ | ✓ | ✓ | ✗ | ✗ |
| HistAuGAN | ✓ | ✓ | ✗ | ✗ |
| ContriMix | ✓ | ✓ | ✓ | ✓ |

In contrast with DRIT++, ContriMix leverages the color difference between random pairs of training samples to train encoders for decoupling the content from the color attribute. Once trained, ContriMix can be used as a stain color augmentation technique to generate synthetic images to train other task-specific networks. See Table 1 for a comparison.

On Camelyon17-WILDS dataset, we demonstrate that backbone networks trained with ContriMix augmentation are capable of achieving color-invariant properties and outperform competing methods in a classification setting. Clustering of ContriMix representations shows that the content encodings are domain-invariant, while the attribute encodings capture color differences across different domains (hospitals). We further perform an in-depth subgroup analysis on slides from the test set and find that backbones trained with ContriMix augmentation have robust performance in presence of tissue patches containing a significant amount of red blood cells, lymphocytes, and low fractional tissue area. We make the source code available for research use, along with ContriMix models trained on Camelyon17-WILDS and 2.5 million images from the Cancer Genome Atlas (TCGA) dataset.

## 2 Method

### 2.1 Model architecture

Figure 1A shows the architecture of ContriMix. It consists of a content encoder $E^c$ that extracts different tissue content such as cell nuclei, connecting tissue etc and an attribute encoder $E^a$ that encodes the color appearance. It also includes an image generator $G$ that takes a content encoding $z^c$ (Figure 1B) and an attribute encoding $z^a$ to generate a synthetic image. The image generator does not need the one-hot encoded domain to generate the output like DRIT++ Lee et al. (2020).

In ContrixMix, all images from the training batch are passed to both encoders to extract the content encodings and attribute encodings. Next, randomly mixed combinations of the content and attribute encodings within the training minibatch are created and fed into the image generator $G$ to create synthetic images. For simplicity, we will use $I_{jk}$ to denote the synthetic image created from the content encoding $z_j^c$ of the $j^{th}$ image, $I_j$, and attribute encoding $z_k^a$ of $I_k$, namely $I_{jk} = G(z_j^c, z_k^a)$, $z_j^c = E^c(I_j)$, and $z_k^a = E^a(I_k)$.

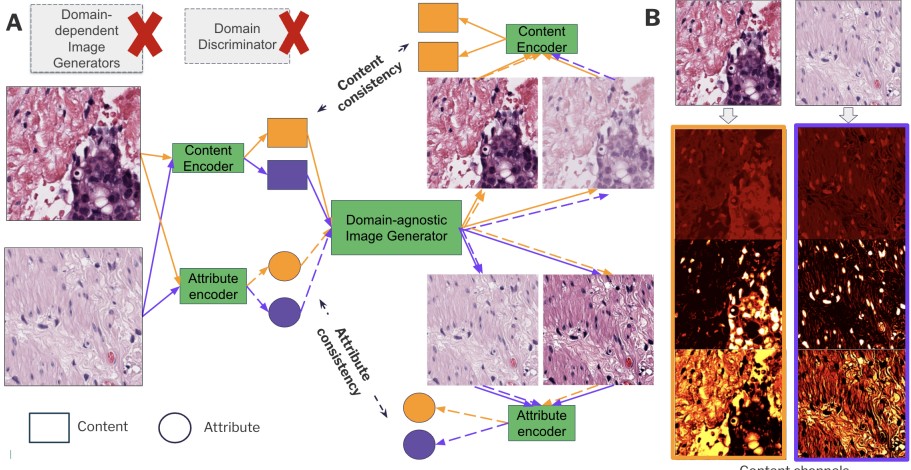

Figure 1: A) Overview of ContriMix - Content and attribute encodings are extracted, randomly mixed, and then combined to generate synthetic images without any domain labels. B) Three example content channels from input images. Different channels highlight different features in the tissue images.

The objective function of ContriMix is

$$L_{ContriMix} = \lambda_a L_{attr.} + \lambda_c L_{cont.} + \lambda_s L_{self-recon.}, \tag{1}$$

where $L_{attr.}, L_{cont.}$, and $L_{self-recon.}$ are the attribute consistency loss, content consistency loss, and the self-reconstruction loss with respective weights $\lambda_a, \lambda_c, \lambda_s$. Note that this objective function is much simpler than that of DRIT++ Lee et al. (2020) which requires adversarial losses for content and domain attribute encodings, latent space reconstruction loss, and KL divergence loss on the attribute encodings to enforce the attribute space to be distributed according to the standard normal distribution. Here, $L_{attr.}$ measures the difference between the extracted attribute of the synthetic image $E^a(I_{jk})$ and that of the self-reconstructed image $E^a(I_{kk})$ to the expected attribute encoding $z_k^a$. $L_{cont.}$ measures the difference between the extracted content from the synthetic image $E^c(I_{jk})$ and that of the self-reconstructed image $E^c(I_{jj})$ to the expected content encoding $z_j^c$. Finally, $L_{self-recon.}$ quantifies the difference between the self-reconstructed images $I_{jj}$ and the input image $I_j$. We use L1 losses in all of these loss terms.

## 2.2 Comparison with other methods

We limit our comparison to other stain color augmentation methods, leaving out those that perform stain normalization to transfer the stain color of the test domain to the training domain. Competing methods include

- Reported methods from https://wilds.stanford.edu/leaderboard/: ERM Sagawa et al. (2021), LISA Yao et al. (2022), IRMX (PAIR Opt) Zhou et al. (2023), and ERM with targeted augmentation Gao et al. (2022).

- HistAuGAN Wagner et al. (2021): We modified ERM training workflow to feed training images into the trained HistAuGAN networks to generate synthetic images with a probability of 0.5. HisAuGAN networks were frozen when training the backbone. The weights of HistAuGAN networks are provided by Wagner et al. (2021). For attribute sampling, we explored two options 1) Sampling with attributes from 3 training domains only and 2) Sampling with attributes from all 5 domains. The first one mimics a practical scenario where no data from the validation set and the test set are available during training, while better performance is expected in the second. We note that the HistAuGAN model from Wagner et al. (2021) was trained on all 5 domains using whole-slide images from Camelyon-17, giving it an advantage in terms of data diversity compared to ContriMix. For each option, we trained the backbone for 40 epochs using the AdamW Loshchilov and Hutter (2017) optimizer with a learning rate of 1e-4 for 5 random seeds.

- Recent methods: STRAP Yamashita et al. (2021) and FuseStyle Khamankar et al. (2023). Due to the lack of publicly available trained models train for Camelyon17-WILDS (STRAP) or source code (FuseStyle), we were unable to compare validation accuracy . The test accuracy comparison is provided in Table 2.

## 3 Training, Results and Discussion

### 3.1 Dataset

The Camelyon17-WILDS dataset contains 450,000 H&E stained 96 x 96 image patches from 5 hospitals. The objective is to classify them to either tumor or normal. The training dataset consists of patches from the first 3 hospitals, while the validation and test datasets are from the $4^{\text{th}}$ and $5^{\text{th}}$ hospitals, see Fig. 2 A.

### 3.2 ContriMix training

To evaluate the effect of ContriMix, we train a DenseNet121 backbone from scratch on the binary classification task using the training split and compared the performance on the out-of-domain test split following the protocol in Koh et al. (2021). We modified the baseline ERM workflow to insert the two ContriMix encoders and image generator between the input and the backbone. We used a weighted sum of the binary cross-entropy loss and the component losses from ContriMix,

$$L_{total} = \lambda_{BCE}L_{BCE} + \lambda_s L_{self-recon.} + \lambda_a L_{attr.} + \lambda_c L_{cont.} \tag{2}$$

where $\lambda_{BCE}$=0.5, $\lambda_s$=0.1, $\lambda_a$=0.1, $\lambda_c$=0.3. We explored different combinations of weights and found that they mainly impact the speed of convergence and not the backbone performance. Moreover, training ContriMix networks separately or jointly with the backbone yielded no significant difference in the backbone performance while joint training was more convenient and slightly faster. Therefore, we used joint training. We used the AdamW Loshchilov and Hutter (2017) optimizer with a learning rate of 1e-4 and an L2-regularization of 1e-4 for all 10 random seeds. Input images are randomly rotated by multiples of 90 degrees, randomly flipped, and passed to ContriMix encoders. The training time was 12 GPU-hours (RTX8000).

Table 2: **Performance comparison on Camelyon17-WILDS**.

| # Method | OOD Val Acc. (%) | Test Acc. (%) |
|---|---|---|
| ERM (rand search) Sagawa et al. (2021) | 85.8 ± 1.9 | 70.8 ± 7.2 |
| HistAuGAN (3 dom.) Wagner et al. (2021) | 85.8 ± 1.1 | 71.4 ± 7.4 |
| IRMX (PAIR Opt) Zhou et al. (2023) | 84.3 ± 1.6 | 74.0 ± 7.2 |
| LISA Yao et al. (2022) | 81.8 ± 1.4 | 77.1 ± 6.9 |
| FuseStyle Khamankar et al. (2023) | - | 90.49 (-) |
| ERM w/ targeted aug Gao et al. (2022) | **92.7 ± 0.7** | 92.1 ± 3.1 |
| HistAuGAN (5 dom.) Wagner et al. (2021) | 87.9 ± 2.3 | 92.6 ± 0.7 |
| STRAP Yamashita et al. (2021) | - | 93.7 ± 0.15 |
| **ContriMix** | **91.9 ± 0.6** | **94.6 ± 1.2** |

Table 3: **Ablation experiments for number of training centers**. We study the impact of dropping entire domains on ContriMix.

| # Train Centers | OOD Val Acc.(%) | Test Acc.(%) |
|---|---|---|
| 3 | 91.9 ± 0.6 | 94.6 ± 1.2 |
| 2 | 87.2 ± 1.3 | 88.8 ± 1.8 |
| 1 | 85.6 ± 1.4 | 86.9 ± 4.0 |

### 3.3 Benchmarking results

Table 2 reports the performance of DenseNet121 backbones trained with different color augmentation methods. ContriMix outperformed other methods in terms of average accuracy on the test set while being second to the ERM with targeted augmentation on the validation set. The test accuracy of ContriMix augmentation is significantly higher than that of HistAuGAN (3-domains augmentation) by 23.2% while being trained on the same data. Interestingly, ContriMix augmentation trained on 3 hospitals also surpasses other augmentation methods trained from data-abundant sources such as HistAuGAN 5-domains and STRAP.

Figure 2B compares the test accuracy of ContriMix against HistAuGAN 3-domains and 5-domains at a WSI level, with error bars denoting the ±1 standard deviation from mean accuracy. We observed significant performance gaps of mean accuracy on slides 23 (37.1%), 28 (29.2%), and 29 (42.3%) between ContriMix and HistAuGAN 3-domains while both are trained using data from the same domains. Upon further inspection, we discovered that the observed gaps were due to the presence of patches with significant amount of red blood cells, patches that are located near tissue margin, or a high number of lymphocytes with dark stained nuclei causing significant color variation. Supplementary section contains examples of such image patches.

To visualize the dependency between the encodings from ContriMix and domains, we pass encodings of 7200 patches to UMAP McInnes et al. (2018). Figure 3 shows that the attribute tensors contain the differences across patches from different domains, while

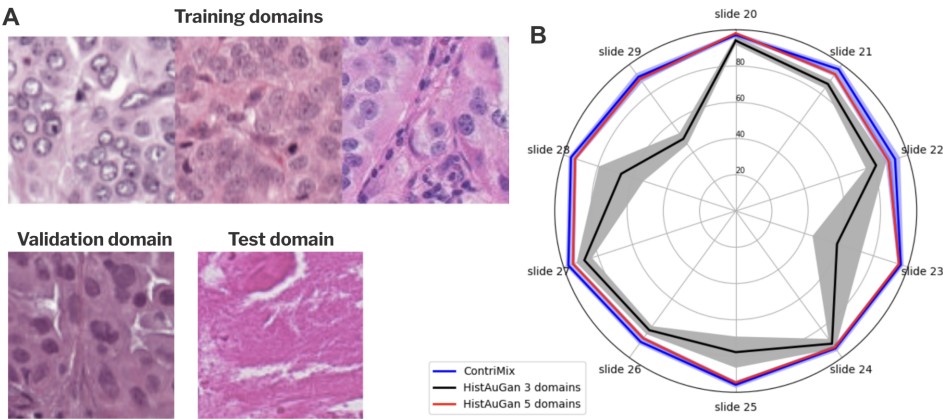

Figure 2: A) Histopathology images from different hospitals in Camelyon-17-WILDS exhibit significant color variation. B) Performance comparison of DenseNet121 backbones trained with ContriMix augmentation, HistAuGAN 3-domains and 5-domains augmentation on 10 different test slides.

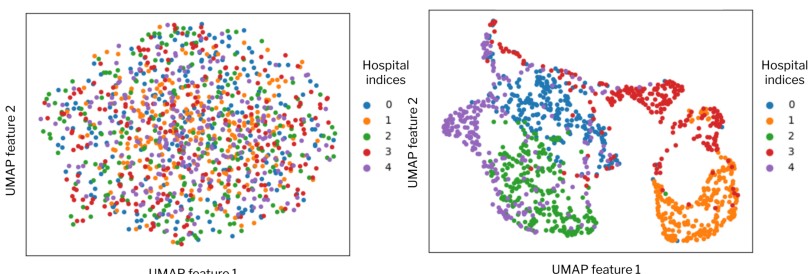

Figure 3: UMAP plot for ContriMix content (left) and attribute (right) encodings colored by different centers in Camelyon17-WILDS.

the content encoder learns center-invariant features. *This happens even though there is no access to domain supervision during training.*

### 3.4 Ablation study - Diversity of training domains

In this ablation, we remove data belonging to different training domains and study its impact on ContriMix. This serves to simulate the real-world setting where we are starved of domain-diverse data. We choose to keep the centers with the least number of samples in the train set - for training with one center, we keep only center 0, while for training with two centers, we keep centers 0 and 3. While there is a drop in performance (Table 3), ContriMix (1 center) is still able to outperform other methods trained on 3 centers. The results indicate that ContriMix is better able to utilize the intra-dataset variations even in the presence of a single domain.

### 3.5 Qualitative evaluation by a board-certified pathologist

We conducted an expert evaluation of the synthetic images generated by ContriMix (Supplementary section). Eighty patches were randomly selected and shared with a board-certified pathologist, with the following question - 'Please evaluate the quality of the synthetic images. Please label the quality as 'NOT SATISFACTORY' if the synthetic image includes any artifact that was not present in the original image, or changes any biological details in the original image'. The pathologist's feedback is as follows - *'All the synthetic images are free of artifacts or changes that would hinder pathologic interpretation'*. The supplementary section additionally contains examples of ContriMix's robustness to image artifacts (ink, blur), ablations around mixing parameters and pseudocode.

## 4 Limitations

At present, there is no systematic way to determine the optimum number of content and attribute channels for ContriMix. A larger than necessary number of attributes may lead to an encoding of redundant information, longer training time but marginal gains in terms of representing true data diversity. In our experiments, a simple hyper-parameter search sufficed, however running this on other image modalities like immunohistochemistry may yield different results.

## 5 Conclusion

In summary, we introduce ContriMix, a scalable technique for stain color augmentation for histopathology images. Through simple mixing of content and attribute within training minibatch along with consistency-based losses, ContriMix can synthesize realistic images with different color appearances while preserving tissue morphology. ContriMix does not require any information about the domain of training patches. This key advantage allows using a trained ContriMix model to extract the stain color (style) from a vast body of unlabeled images and use them to further increase the diversity of synthetic images for color augmentation. We demonstrate that backbones trained with ContriMix color augmentation have a better out-of-domain accuracy compared to other color augmentation methods on Camelyon WILDS-17. Our ablation studies suggest the effectiveness of ContriMix for generating domain-invariant representations without needing domain labels, along with desirable properties such as robustness to color variation from rare substances and learning meaningful representations even in data-diversity starved regimes. We release our code and trained models for research use.

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
