# OpenReview forum: "ContriMix: Scalable stain color augmentation for domain generalization without domain labels in digital pathology"
_MICCAI.org/2024/Workshop/COMPAYL — COMPAYL 2024_

### Official Review · Reviewer_USy4 · 2024-07-05
**In this study the authors focus on a scalable stain color augmentation method for domain generalization without domain labels in the context of digital pathology.**

**Custom Rating:** 5
**Confidence:** 5

**Review:**

The study includes extensive experiments to determine optimal parameters and settings for training and validating the model, aiming to reduce performance variability in stain-heterogeneous data for the task of tumor versus normal classification. However, the following questions need to be addressed to improve the manuscript:


1.	At the beginning of the introduction, the authors mentioned that “color variations lead to poor algorithm performance”. To support this statement, it would be valuable to provide some examples work showing that variability of scanners or patches (domains) affects algorithm performance.
2.	How would the authors see those contributions such as domain adversarial training and stain mix-up-based training, do they not eliminate the influence of stain color variability and help create more generalized models? Was any comparison performed?
3.	How do the authors view model performance on questionable quality problems such as tissue folds and thick tissue regions? Were these issues considered?
4.	Did the authors conduct experiments to observe the robustness of the algorithm by training it on one centre and testing it with four remaining centres, or vice versa? Or was there an experiment to evaluate the within-centre variations and observe the generalizability of the model on a test set with more variances?
5.	How does the inference mechanism work and what is the overhead cost per patch, and has it been compared to other methods?
6.	What is the contribution of content encoder in the overall performance of the model when looking at the provided UMAP representation of data (in Figure 3) shows clear differences in stains but higher randomness in contents?

---

### Official Review · Reviewer_UHPa · 2024-07-08
**A novel approach to generative AI in H&E histopathology**

**Custom Rating:** 4
**Confidence:** 3

**Review:**

The paper nicely describes the design and implementation of a novel approach to generate high quality histopathology images, using a novel stain and morphology disentanglement method. While the method has been successfully benchmarked against a variety of other methods using out-of-distribution accuracy, the implications are not obvious, in particular with respect to clinical application scenarios, such as the number of annotations we can save in a fully supervised setting of we employ the proposed method. Second the discussion does not elaborate on how this approach could be used in situations which require the quantification of stain to derive biological information.

---

### Official Review · Reviewer_cPAM · 2024-07-10
**The paper introduces ContriMix, a novel technique for stain color augmentation in digital pathology images designed to improve the robustness and generalizability of deep learning models. This method addresses the issue of color variability in histopathology images caused by differences in staining and imaging procedures, which can lead to poor model performance when deployed on datasets from sources different from the training set. ContriMix is built upon the DRIT++ framework, a style transfer method that disentangles content and style representations, allowing for scalable stain color augmentation without requiring domain labels.**

**Custom Rating:** 4
**Confidence:** 5

**Review:**

Overall, ContriMix presents a promising approach to stain color augmentation that enhances the robustness and generalizability of deep learning models in digital pathology. Further validation across diverse datasets and comparison with alternative techniques would solidify its position as a leading method in this field.
# Strengths
* Domain Label Independence.
* Scalability and Flexibility
* Preservation of Tissue Morphology
* Robustness to Rare Substances（robust to color variations caused by rare substances）
* Learning Meaningful Representations
# Weaknesses and Areas for Improvement:
* Limitations of Mixing Strategies. (The paper investigates the impact of the number of mixes and random versus targeted mixing but could provide deeper insight into how these strategies affect the quality of augmentation and model performance.)
* Generalizability: Although ContriMix is demonstrated to work well on C17-WILDS dataset, its applicability to other datasets remains to be explored.
* Release of Code and Models

# some questions
* I think this paper is kind of "style transfer" work,disenangling content and style is very common in this field, what's new of this paper?
* As the development of pathology foundation model in computational pathology (CPath), are the synthetic data  stil necessary? What do you think the future of the DG in CPath?
* For example, could foundation model feature extractor + linear prob (or finetune) defeat DG methd?

---

### Decision · Program_Chairs · 2024-07-16

Accept